# Toward Interpretable Evaluation Measures for Time Series Segmentation

**Félix Chavelli**
Inria, ENS, CNRS, PSL
Paris, France
felix.chavelli@inria.fr

**Paul Boniol**
Inria, ENS, CNRS, PSL
Paris, France
paul.boniol@inria.fr

**Michaël Thomazo**
Inria, ENS, CNRS, PSL
Paris, France
michael.thomazo@inria.fr

## Abstract

Time series segmentation is a fundamental task in analyzing temporal data across various domains, from human activity recognition to energy monitoring. While numerous state-of-the-art methods have been developed to tackle this problem, the evaluation of their performance remains critically limited. Existing measures predominantly focus on change point accuracy or rely on point-based measures such as Adjusted Rand Index (ARI), which fail to capture the quality of the detected segments, ignore the nature of errors, and offer limited interpretability. In this paper, we address these shortcomings by introducing two novel evaluation measures: **WARI** (Weighted Adjusted Rand Index), that accounts for the position of segmentation errors, and **SMS** (State Matching Score), a fine-grained measure that identifies and scores four fundamental types of segmentation errors while allowing error-specific weighting. We empirically validate WARI and SMS on synthetic and real-world benchmarks, showing that they not only provide a more accurate assessment of segmentation quality but also uncover insights, such as error provenance and type, that are inaccessible with traditional measures.

## 1 Introduction

Massive collections of time-varying measurements, commonly referred to as *time series*, have become a reality in every scientific and industrial domain [1–4]. Such temporal measurements can correspond to different physical quantities, such as temperature and pressure [5], electricity consumption [6], or human pose [7]. Several tasks have emerged from the pressing need to analyze time series, such as classification [8], clustering [9], anomaly detection [10–12], motif discovery [13], and time series segmentation [14]. The latter (sometimes referred to as *change point detection* or *state detection*) is a crucial task. The goal is to identify distinct states or patterns within the data, which can provide valuable insights into the underlying processes. More generally, time series segmentation aims to respectively detect change points, which delineate different states, and to cluster these states in order to recognize recurring states. Such states can correspond to a human activity such as *walking* and *running* [15], or specific appliances in electrical consumption time series [6]. Although a wide variety of state-of-the-art algorithms have been proposed, leveraging diverse approaches (e.g., statistical methods [14, 16, 17], Markov models [18, 19], auto-encoders [20–22], or symbolic representations [23]), enabling notable progress in recent benchmarks, we observe three major limitations that undermine their ability to reliably assess segmentation quality.

First, change point-based measures, which focus solely on the accuracy of detecting transition points, **do not adequately capture the overall quality of the segmentation** itself. Even if change points are correctly detected, the resulting segment labels might still be incorrect or uninformative. Second, most widely used measures, such as Adjusted Rand Index (ARI), are point-based and thus treat all errors (i.e., points belonging to wrongly segmented subsequences) equally, **failing to distinguish**

**between different types of errors**. For instance, a delay in detecting a transition may simply reflect minor misalignment with human annotation, whereas an isolated error, such as labeling an entire segment incorrectly, is far more severe. These two types of errors carry very different implications, yet traditional measures weight them equally. Lastly, current evaluation measures cannot track and categorize the nature of errors (e.g., delay vs. isolation), leading to **limited interpretability**. This hinders deeper diagnostic and reduces the practical value of the measure for improving models.

To address these limitations, we introduce two new evaluation measures: **WARI** (**W**eighted **A**djusted **R**and **I**ndex) and **SMS** (**S**tate **M**atching **S**core). The first measure, WARI, extends the traditional Adjusted Rand Index by incorporating the temporal position of segmentation errors. This allows it to differentiate between positions of errors, such as between errors close but misaligned with the ground truth, and isolated errors which indicate more substantial segmentation failures. While WARI provides a more nuanced and temporally aware variant of ARI, SMS offers a complementary perspective by explicitly identifying and scoring four fundamental types of segmentation errors. More specifically, SMS allows practitioners to weight each error type based on their application, thus providing a customizable and interpretable evaluation measure. By maintaining error provenance and enabling targeted analysis, SMS enhances the transparency of segmentation assessments. Finally, we empirically evaluate the validity of WARI and SMS by comparing them to existing measures. We then report the impact of our measures on the evaluation of state-of-the-art segmentation methods. We also provide additional insights, such as the prevalence and severity of specific error types, that were previously inaccessible. Overall, our contributions are as follows:

- We provide a thorough analysis of the literature on time series segmentation and propose a typology of fundamental and distinct segmentation errors (cf. **Sec. 2.1** to **2.2**).

- We critically examine the limitations of existing evaluation measures, highlighting their inability to capture key aspects of segmentation quality (cf. **Sec. 2.3**).

- We introduce our two novel evaluation measures, WARI and SMS, and provide detailed descriptions of their design, objectives, and theoretical advantages (cf. **Sec. 3**).

- We empirically demonstrate that our proposed measures offer a more appropriate and insightful evaluation of segmentation quality compared to existing measures (cf. **Sec. 4.1**).

- We analyze WARI and SMS impact on the assessment of state-of-the-art segmentation methods, and present novel insights that are impossible with traditional evaluation measures (cf. **Sec. 4.2**).

## 2   Background and Foundations

This section provides the foundational concepts and formal definitions necessary for understanding time series segmentation and evaluation strategies. We first introduce fundamental definitions to assess the technical differences between problem formulations and existing algorithms.

**Definition 1 (Real-valued Time Series)** *A real-valued time series of length $N$ and dimension $D$ is a time-ordered sequence denoted by $T = [t_1, \ldots, t_N]$, where each $t_i \in \mathbb{R}^D$ for $i = 1, \ldots, N$.*

We define a univariate time series as a time series with $D = 1$. Moreover, a subsequence of $T$ from index $i$ to $j$ (with $1 \leq i \leq j \leq N$) is denoted by $T_{[i,j]} = [t_i, t_{i+1}, \ldots, t_j]$ and has length $j - i + 1$. We now define a state sequence as follows:

**Definition 2 (State Sequence)** *A state sequence $S = [s_1, \ldots, s_N]$ associated with a time series $T = [t_1, \ldots, t_N]$ is a sequence of the same length, where each $s_i \in \mathcal{S}$ is a discrete label representing the latent state of the system at time step $i$, and $\mathcal{S}$ is a finite set of possible states. In a state sequence, $i$ (with $1 \leq i < N$) is a **change point** if $s_i \neq s_{i+1}$.*

### 2.1   Time Series Segmentation: A Multifaceted Problem

Time series segmentation refers to the task of dividing a time series into meaningful and homogeneous segments, where each segment corresponds to a period during which the underlying generative process is assumed to be stable. Two common approaches to segmentation are **change point detection** and **state detection**, illustrated in Fig. 1. While they differ in assumptions and goals, both aim to capture shifts in the behavior in the time series.

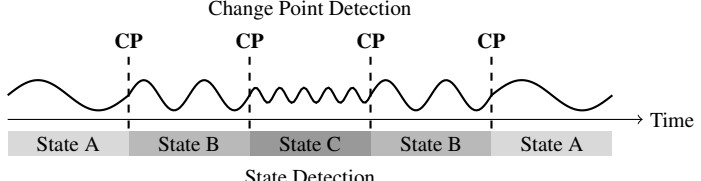

Figure 1: Illustration of Change Point Detection vs. State Detection.

### 2.1.1 Change Point Detection

The output of a change point detection algorithm is an increasing sequence of integers $(c_1, \ldots, c_M)$, where each change point (CP) marks a transition between two segments. The resulting segments are contiguous and non-overlapping, each corresponding to a stationary or stable regime. Change point detection is typically used when the goal is to precisely localize transitions.

A large panel of approaches tackling change point detection has been introduced in the literature. First, profile-based methods, such as ClaSP [14], FLUSS [16], and ESPRESSO [17], typically operate by constructing a profile from the time series and identifying CPs at local extrema. Second, other methods proposed in the literature rely on statistical principles. For instance, binary segmentation [24] (BinSeg) employs recursive likelihood hypothesis testing, and PELT [25] offers a pruned, optimization-based variant. Finally, Bayesian approaches, such as BOCD [26], sequentially update the probability of a CP presence as new data arrives.

### 2.1.2 State Detection

In contrast, state detection assumes that the time series is generated by an underlying sequence of latent states. Each state corresponds to a specific pattern or regime, and the same state can recur at different points in time. Hence, the output of a state detection algorithm is a predicted state sequence $P = (p_1, \ldots, p_N)$. The primary goal of state detection is to identify changes in the latent state itself, allowing for the recognition of recurring patterns, rather than simply detecting any statistical shift.

Diverse approaches have been proposed for state detection. These include methods based on encoder architectures (e.g., E2USD [20], Time2State [21], HVGH [22]), convolutional neural networks such as RP-mask [27] and PrecTime [28], graph representations such as GRAB [29] and uGLAD [30]), probabilistic graphical models such as hidden Markov models (e.g., HDP-HSMM [18] and MASA [19]), Markov random fields (e.g., TICC [31]), or rule-based systems (e.g., PaTSS [23]).

Importantly, change point detection can be viewed as a subproblem of state detection. Once change points have been identified, they partition the time series into segments. A clustering algorithm can then be applied to these segments to assign state labels (as performed in [20, 21] for the change point detection method ClaSP [14]). This two-step approach enables the reconstruction of a state sequence from raw change points, highlighting that state detection generalizes change point detection. Thus, for the rest of the paper, we will mainly focus on the state detection problem.

### 2.2 Evaluating Time Series Segmentation: Typology of Errors and Desired Properties

In order to accurately assess the quality of a segmentation when compared with a ground-truth, we need to formally define segmentation error types. Let $\mathcal{S} = \{s_1, \ldots, s_M\}$ be a finite set of states. Let $R = (r_1, r_2, \ldots, r_N) \in \mathcal{S}^N$ be the *real*, state sequence, and let $P = (p_1, p_2, \ldots, p_N) \in \mathcal{S}^N$ the *predicted* state sequence. We define an *error block* as a maximal contiguous index interval $[i, j]$ that cannot be extended without including a correctly classified point such that $\forall k, l \in [i, j], \ p_k = p_l$ and $\forall k \in [i, j], \ p_k \neq r_k$.

For an error block $[i, j]$ in $P$, we define the *atomicity* $A_{[i,j]} = \left| \{ r_k : k \in [i, j] \} \right|$ as the number of distinct states within $R_{[i,j]}$. Based on $A_{[i,j]}$, we introduce a novel typology of errors (illustrated in Fig. 2), such that each error block belongs to exactly one error type. Our typology is as follows:

**Delay** ($A = 1$): The real and predicted states within $[i, j]$ are constant, say $r$ and $p$ respectively. Moreover, at least one block neighbor exists and satisfies $r_{i-1} = p_{i-1} = p$ or $r_{j+1} = p_{j+1} = p$.

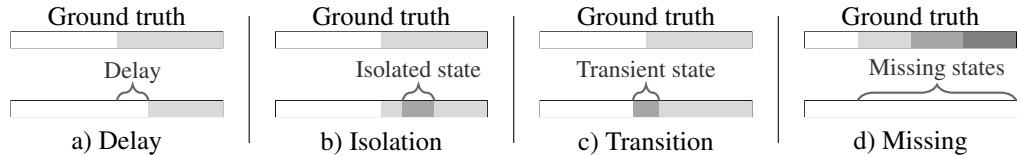

Figure 2: Ground truth (top) with four error examples below: delay, isolation, transition, and missing.

**Isolation** ($A = 1$): The real and predicted states within $[i, j]$ are constant, say $r$ and $p$ respectively. Moreover, the error occurs in a middle of a constant real state, when $r_{i-1} = r_{j+1} = r$.

**Transition** ($A = 2$): There are exactly two distinct real states within $[i, j]$, indicating the error spans a real state change.

**Missing** ($A \geq 3$): Three or more distinct real states appear within $[i, j]$, indicating omission of three or more real states.

This typology is important as some errors might be less severe than others. Indeed, state boundaries are generally identified in a strictly binary manner, which may not suit some real-world applications. In practice, transitions between activities are often gradual rather than instantaneous. Thus, differences between real and predicted states can arise from alternative interpretations of these transient periods. While some research propose to introduce gradual labelling [23], a simpler strategy would be to propose measures that are robust to labeling ambiguities—for instance, deciding exactly where to place the boundary between walking and running, with the hypothesis that in such cases, an error near a real boundary (i.e., transient state or delay) is less severe than one that occurs in the middle of a homogeneous region (i.e., missing state or isolated state).

**Desired Properties:** Based on the need to rank error types (as motivated above), we propose a set of properties that should be satisfied by any measure for state detection. These properties are designed to provide a more meaningful evaluation of state detection algorithms.

**P1**: *The measure should be sensitive to the errors **length**, with larger errors leading to lower scores.*

**P2**: *The measure should account for the temporal structure, penalizing **positions** of errors differently.*

**P3**: *The measure should be sensitive to the **type** of error, with different penalties for different types.*

**P4**: *The measure should be **interpretable** and provide insights into the quality of the segmentation.*

While these properties provide valuable guidance for the development and evaluation of state detection measures, we emphasize that they are not formal axioms and may not be strictly or simultaneously satisfied in all cases. For example, although **Property P1** suggests that longer errors should lead to lower scores, the impact of an error's length may depend on its context, such as whether it results from a *delay* or from an *isolated* error. In such cases, the score may be moderated by considerations of temporal position (**Property P2**) or error type (**Property P3**). Thus, these properties should be viewed as guiding principles rather than rigid requirements.

## 2.3 Existing Measures and Limitations

While several measures have been adopted in the literature, each comes with a set of assumptions and drawbacks, failing to catch some of the desired properties mentioned above. This section reviews these commonly used measures, discussing their strengths and limitations.

### 2.3.1 Change Point Detection Measures

Among the commonly used measures for change point detection, the **F1 score** is a harmonic measure that combines precision and recall. Following previous work, we value a margin tolerant F1 score, identifying the correct detections by matching predicted change points to ground-truth annotations within a given *margin*, while preventing double-counting of predictions by removing matched points after association. However, selecting the appropriate margin is challenging. The example in Fig. 3(a) illustrates two scenarios S1 and S2 in which the F1 score is 1, although S1 contains a longer error than S2, thus failing to meet **Property P1**. A margin parameter that is a function of the time series length is often preferred, as proposed in [14], where it is set to 1% of the time series length.

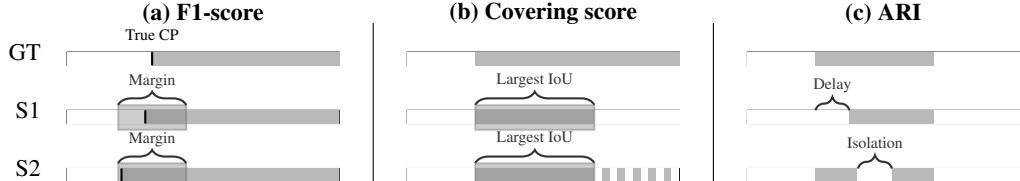

Figure 3: Limitations of (a) F1, (b) Covering, and (c) ARI scores. For two different segmentations (S1 more accurate than S2 according to the ground truth GT), all measures return the same score.

The **covering score**, another commonly used measure, captures segment-level similarity rather than exact change point matching. Unlike the F1 score, which treats change points as discrete events, covering accounts for segment overlap. It is defined as the average of the intersection over union (IoU) scores for each segment in the ground truth, normalized by the number of segments. With $R$ and $P$ representing real and predicted state sequences, the Covering score is calculated as follows:

$$C = \frac{1}{N} \sum_{r \in R} |r| \max_{p \in P} \frac{|r \cap p|}{|r \cup p|} \tag{1}$$

However, the covering score can assign identical scores to segmentations that are qualitatively different, thus, failing to meet **Property P1**. Fig. 3(b) illustrates this limitation with two predicted segmentations that achieve the same covering score despite significant differences. This highlights the need for complementary measures that better capture segmentation quality.

### 2.3.2 State Detection Measures

State detection performance is most commonly evaluated with clustering-based measures [21, 20], such as the **Adjusted Rand Index** (ARI), the **Normalized Mutual Information** (NMI) and the **Adjusted Mutual Information** (AMI). In the rest of the paper, we will focus on ARI and we provide details on the additional clustering-based measures in the Appendix.

The computation of ARI is based on the Rand Index (RI) that computes the fraction of agreeing pairs (i.e., pairs that are either grouped or separated together) over the total number of pairs. Formally, with $R$ and $P$ representing real and predicted state sequences and $U_R = \{r_i : r_i \in R\}$ and $U_P = \{p_i : p_i \in P\}$ the unique sets of states, we define the *contingency matrix* $C = [n_{ij}]$ of size $|U_R| \times |U_P|$ with $n_{ij} = \sum_{k=1}^{N} \mathbf{1}\{r_k = U_R[i] \wedge p_k = U_P[j]\}$, i.e., the number of observations at timestamp $k$ that belong to state $U_R[i]$ in the first state sequence $R$ and $U_P[j]$ in the second state sequence $P$. Finally, with $\mathbb{E}[\text{RI}]$ the expected Rand Index under a random model, ARI is computed as follows:

$$\text{RI} = \frac{\sum_{i,j} \binom{n_{ij}}{2}}{\binom{N}{2}}, \quad \text{ARI} = \frac{\text{RI} - \mathbb{E}[\text{RI}]}{1 - \mathbb{E}[\text{RI}]} \tag{2}$$

As shown in the equations above, the Adjusted Rand Index (ARI) is sensitive to the number of matching temporal point pairs between segmentations. Thus, the total number of segmentation errors directly influences the ARI score, satisfying **Property P1**. However, clustering-based measures like ARI are inherently point-based and do not account for the position or type of segmentation errors. For example, Fig. 3(c) demonstrates this limitation: two predicted segmentations yield the same ARI score despite exhibiting markedly different segmentation error patterns (one delay versus one isolated error). Consequently, clustering-based measures fail to satisfy **Properties P2** and **P3**.

## 3 WARI and SMS: Our Proposed Measures

As outlined in the previous section, existing evaluation measures exhibit significant limitations (failing to meet either **Property P1**, **P2** or **P3**). Moreover, these measures provide limited interpretability, making it difficult for practitioners to understand accuracy scores or pinpoint specific weaknesses and areas for improvement (failing to meet **Property P4**). To address these shortcomings, we propose

two state detection measures. The first one, **WARI**, consists of a modified (*weighted*) version of the standard ARI, making it *distance-to-boundary* sensitive. The second one is a new measure, namely **SMS** (**S**tate **M**atching **S**core), that identifies a mapping between the predicted and real states, which is then used to compute a score based on the contexts and types of errors encountered according to the taxonomy defined in the previous section (Sec. 2.2).

### 3.1 Toward Position-Sensitivity: The Weighted Adjusted Rand Index

As mentioned in Sec. 2.3, the Adjusted Rand Index (ARI) treats all segmentation errors equally. However, segmentation errors near cluster boundaries are less critical than errors in the cluster interior (i.e., **Property P2**). To account for this, we define a weighted version of ARI, *Weighted Adjusted Rand Index* (WARI), based on the distances to change points. More specifically, this distance, called $d_i$, is defined for each time step $i$ and corresponds to the distance from the nearest ground truth change point. We then define a weight $w_i = 1 + \alpha\, d_i$ for each timestamp. $\alpha \geq 0$ is a user-parameter, set by default to 0.1 in the rest of the paper. For $\alpha > 0$, observations deep inside ground truth segments (i.e., with high $d_i$) are given more weight, and thus, more penalized if wrongly predicted.

**Weighted Contingency Matrix:** In the weighted setting described above, the contingency matrix (defined in Sec. 2.3.2) is adapted by replacing counts with weighted sums: $\widetilde{n}_{ij} = \sum_{x_k \in U_i \cap V_j} w_k$. The weighted Adjusted Rand Index is then computed by using $\widetilde{n}_{ij}$ values (and the corresponding total weighted sum of pairs) in Equation 2. Note that such weighted procedure can be applied to other clustering-based measures, such as Normalized Mutual Information (NMI) and Adjusted Mutual Information (AMI). We provide more details in the Appendix.

**Properties:** WARI behaves like ARI with a boundary-aware lens. When $\alpha = 0$, the weights collapse to $w_i \equiv 1$ and WARI exactly coincides with ARI. As soon as $\alpha > 0$, the measure starts to "prefer" boundary-adjacent mistakes: points far from change points receive larger weights than those near them, so interior misclassifications are penalized more strongly, encoding the desired position sensitivity (P2). Yet the overall scale remains familiar—WARI reaches 1 under perfect agreement, drifts toward 0 for random labelings, and may become negative for strongly discordant segmentations—preserving ARI's qualitative range and interpretation.

**Sensitivity to the position parameter $\alpha$:** Because $w_i(\alpha) = 1 + \alpha\, d_i$ varies linearly in $\alpha$ with $d_i \in [0, D_{\max}]$, the weighted contingency and the resulting WARI vary smoothly with $\alpha$. In particular, for two settings $\alpha$ and $\alpha'$, the score difference is bounded linearly: $|\mathrm{WARI}(\alpha) - \mathrm{WARI}(\alpha')| \leq L\,|\alpha - \alpha'|$, where $L$ depends only on the dataset via distances to boundaries and segment masses (details in Appendix). Practically, this yields a *robust* behavior around the default $\alpha{=}0.1$: moderate changes of $\alpha$ produce limited, predictable variations in the score. Larger $\alpha$ emphasizes interior purity (harsher penalties far from boundaries), while smaller $\alpha$ increases boundary tolerance.

---

**Algorithm 1** Optimal State Mapping

---

**Require:** The real and predicted state sequences $R = (r_1, \ldots, r_N)$ and $P = (p_1, \ldots, p_N)$.
1: Compute **unique sets** $U_R = \{r_i : r_i \in R\}$ and $U_P = \{p_i : p_i \in P\}$
2: Compute **cost matrix** $C$ of size $|U_P| \times |U_R|$, such that, for $p_u \in U_P$ (row $i$) and $r_u \in U_R$ (column $j$), the negative overlap $C_{ij}$ is as follows:

$$C_{ij} = -\sum_{k=1}^{N} \mathbf{1}\{p_k = p_u \wedge r_k = r_u\}.$$

3: Find **Optimal Assignment:** Apply the Hungarian algorithm [32] to $C$ to find a mapping $\mathcal{M}$ from states in $U_P$ to states in $U_R$ that minimizes the total cost (maximizes total overlap).
4: **for all** $p_u \in U_P$ not assigned by $\mathcal{M}$ **do**                    ▷ Handle unassigned predicted labels
5:    Set $\mathcal{M}(p_u)$ to the smallest non-negative integer $m$ such that $m$ is not assigned by $\mathcal{M}$.
6: **end for**
7: **return** Final mapping $\mathcal{M}$.

---

---

**Algorithm 2** State Matching Score (SMS)

---

**Require:** Real sequence $R$, prediction $P$, mapping $\mathcal{M}$, penalty weight $w = \{w_{\text{delay}}, w_{\text{transition}}, w_{\text{isolation}}, w_{\text{missing}}\}$

1: $\tilde{P} \leftarrow (\mathcal{M}(p_1), \ldots, \mathcal{M}(p_N))$ ▷ Map predictions to real states
2: Let $\mathcal{B}$ be the set of error blocks in $\tilde{P}$ (cf. typology, Sec. 2.2).
3: **for all** $b = [i, j] \in \mathcal{B}$ **do**
4:     $l \leftarrow j - i + 1$    (block length)
5:     $A \leftarrow |\{r_k : i \leq k \leq j\}|$   (atomicity)
6:     $e \leftarrow \{\text{delay}, \text{isolation}, \text{transition}, \text{missing}\}$    (determine error type)
7:     **if** $e \in \{\text{isolation}, \text{transition}\}$ **then**
8:         find nearest real change points $b_{\text{prev}} < i$ and $b_{\text{next}} > j$
9:         $d \leftarrow \dfrac{2 \min(i - b_{\text{prev}},\ b_{\text{next}} - j)}{N}$   (normalized distance to change point)
10:     **end if**

11:     $\text{Pen}(b) \ = \ \begin{cases} l\,(1 + w_e), & e = \text{delay}, \\ l\big(1 + d\,w_e\big), & e \in \{\text{isolation}, \text{transition}\}, \\ l\Big(1 + w_e\,(1 + \frac{3}{A}(w_e - 1))\Big), & e = \text{missing}. \end{cases}$

12: **end for**
13: **return** $\text{SMS} = 1 - \dfrac{1}{N} \sum_{b \in \mathcal{B}} \text{Pen}(b)$.

---

## 3.2 Enhancing Interpretability: The State Matching Score

Whereas WARI takes into account the position of the errors (i.e., satisfying **Property P2**), the types of the errors are not considered in the accuracy score, and the interpretability of the score is low (i.e., failing to meet **Properties P3** and **P4**). To address these shortcomings, We introduce a novel interpretable and customizable measure, called the State Matching Score (SMS). The core idea relies on aligning the predicted and ground truth state sequences, taking into account the types and associated severity of errors made by the algorithm.

The State Matching Score (SMS) is computed through a two-stage process, detailed in Algorithm 1 and 2. First, an optimal mapping between predicted and real states is established using the Hungarian algorithm [32] on a cost matrix representing the negative overlap between unique states (Algorithm 1). This ensures that predicted state labels are aligned with real state labels in a way that maximizes overall agreement. Second, the State Matching Score itself is computed (Algorithm 2). This involves identifying error blocks in the mapped predicted sequence, classifying these errors according to the typology in Sec. 2.2, and assigning a penalty to each block based on its type, length, and context (i.e. distance to real boundaries, atomicity). The final SMS is a normalized score reflecting the overall quality of the state detection.

As shown in Algorithm 2, SMS incorporates penalty weights for different error types, allowing for customization to specific applications. For instance, in scenarios where reaction time is critical and false positives are tolerable, delays might be penalized more heavily than missing states. Despite this flexibility, the SMS exhibits robustness to the choice of these penalty weights. The overall score is primarily influenced by the total number of errors rather than the precise weight distribution.

**Formal properties and guarantees.** SMS is designed to be interpretable, predictable and robust to changes of its knobs. Intuitively, the score first reflects how much time is mislabelled (the total length of error blocks), and then applies controlled, interpretable refinements for error types and contexts.

If all penalty weights are set to zero, SMS simply reduces to the fraction of correctly labelled time:

$$\text{SMS} = 1 - \frac{E}{N},$$

where $E$ is the total length of error blocks and $N$ the sequence length. With nonzero weights, since $d \in [0, 1]$ and $A \geq 3$ (as defined above), the score remains tightly bounded:

$$1 - \frac{(1 + w_{\max})\,E}{N} \ \leq \ \text{SMS} \ \leq \ 1 - \frac{E}{N},$$

where $w_{\max}$ is the largest penalty weight. Thus, weights can only modulate the score within explicit, predictable limits around the baseline $1 - E/N$.

Changing the weights moderately cannot swing the score wildly. For two weight settings $w$ and $w'$, the difference in scores is bounded by

$$|\text{SMS}(w) - \text{SMS}(w')| \leq \|w - w'\|_\infty \frac{E}{N}.$$

When the overall error mass $E/N$ is small—as desired for good segmentations—SMS is provably stable to weight choices. We provide additional experiments measuring the robustness of SMS in the Appendix.

## 4 Experimental Evaluation

We now empirically evaluate the advantages of our proposed measures. In total, we consider a panel of 6 segmentation methods (E2USD [20], Time2State [21], HDP-HSMM [18], TICC [31], ClaSP [14] (used with kMeans clustering), and PaTSS [23]). We exclude HVGH [22] due to its poor performance [20, 21], and AutoPlait [33], which previous studies reported as non-functional [20, 21]. In addition, we consider a benchmark of 5 datasets (PAMAP2 [15], USC-HAD [34], UCR-SEG [35], ActRecTut [36], MoCap [37]) spanning various domains. Given the emphasis of this paper on state detection, particular attention is given to comparing WARI and SMS with ARI, i.e., the most commonly used measure in the literature. Finally, we provide an open-source implementation [1] of our measures and evaluation. Additional experimental setup details can be found in Appendix.

### 4.1 Evaluating the Evaluation Measures

We first design a synthetic experiment that evaluates sensitivity to error **length**, **position**, and **type**. The results, shown in Fig. 4, highlight distinct behaviors across the three measures (ARI, WARI, and SMS). First, all measures are sensitive to the error length (cf. Fig. 4(a)), exhibiting decreasing scores as segmentation errors grow is length, across segmentations S1 to S9 (satisfying **Property 1**). Second, while WARI and SMS react to the **position** of the error—penalizing isolated errors more heavily—ARI remains insensitive (cf. Fig. 4(b)), assigning a constant score regardless of the error's location (failing to meet **Property 2**). Finally, we assess sensitivity to **error type**, comparing measures behavior on a delay and a transition error of same lenght (cf. Fig. 4(c)). While SMS assigns different scores to each case, both ARI and WARI return identical values, demonstrating their insensitivity to error type (failing to meet **Property 3**).

We now present in Fig. 5 a qualitative comparison of segmentation results from E2USD and Time2State on a MoCap dataset time series. Traditional measures like ARI marginally favor E2USD, despite exhibiting clear isolated errors and a less accurate segmentation overall. In contrast, Time2State produces a more consistent segmentation, primarily with delay and transition errors. The proposed SMS, along with WARI, pick Time2State's output as the best segmentation. Specifically, SMS offers an interpretable diagnostic of error types, a feature lacking in conventional measures.

---

[1] Public Repository: `https://github.com/fchavelli/tsseg-eval/`

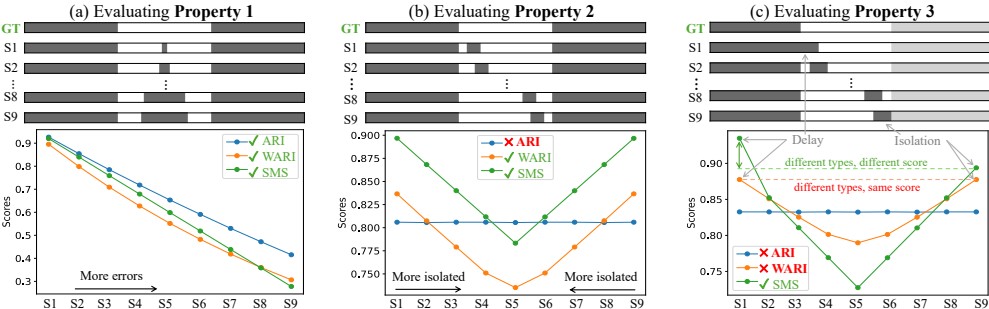

Figure 4: Synthetic data examples illustrating various error types and measure responses.

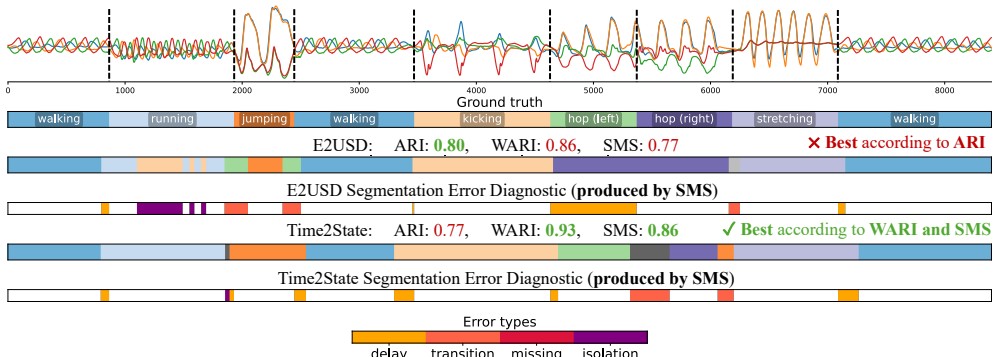

Figure 5: Segmentation of a time series from the MoCap dataset using E2USD and Time2State.

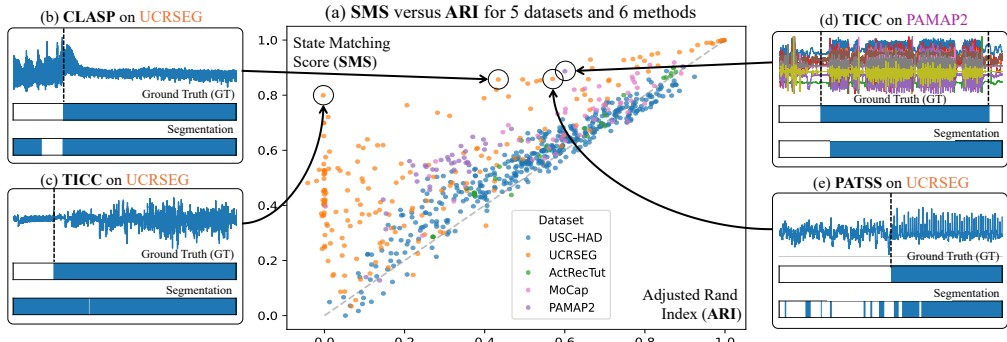

Figure 6: Comparison of ARI and SMS on real datasets collection.

More generally, we investigate how the proposed WARI and SMS measures compare to ARI on real-world data. Fig. 6 shows a scatter plot of SMS versus ARI across the 5 datasets with the 6 segmentation methods (the comparison with WARI can be found in Appendix). While most points align along the diagonal (i.e., similar ARI and SMS scores), several points deviate significantly. In many cases, these deviations arise in settings with very few ground truth segments, where ARI assigns low scores for predictions consisting of a single segment. In contrast, SMS still assigns a proportional score based on how much of the smallest ground truth segment is recovered. For example, Fig. 6(c) shows an time series from the UCRSEG dataset where TICC predicts a single segment, resulting in ARI $\approx 0$, whereas SMS captures the partial match and yields a higher score. In other scenarios, such as Fig 6(b) and 6(e), SMS assigns more favorable scores than ARI due to its tolerance to temporal misalignment (e.g., delay errors), especially in cases where ground truth labels may themselves be subjective or ambiguous. This illustrates that SMS can better capture meaningful segmentations despite imperfect annotations. However, as SMS is not adjusted for chance, it may overvalue simplistic segmentations (e.g., a single segment) where error types are less relevant due to few states. As WARI (shown to be more accurate than ARI earlier) is adjusted for chance, we highlight the complementary nature of SMS and WARI, suggesting the interest in their joint usage.

## 4.2 Impact on State of the Art

We evaluate the relative performance of segmentation algorithms across datasets, using the pairwise Wilcoxon sign rank test, with a critical value of $\alpha = 0.05$. Each time series is treated as an individual test instance. The corresponding critical diagrams are in Fig. 7 for ARI, WARI and SMS. Critical Diagrams for F1 and Covering can be found in the Appendix.

**Where we are:** The algorithm rankings remain consistent, with the only notable exception being SMS on univariate datasets. In the univariate context, ClaSP systematically achieves the highest rank, aligning with previous works results [20, 21]. Regarding multivariate time series, Time2State

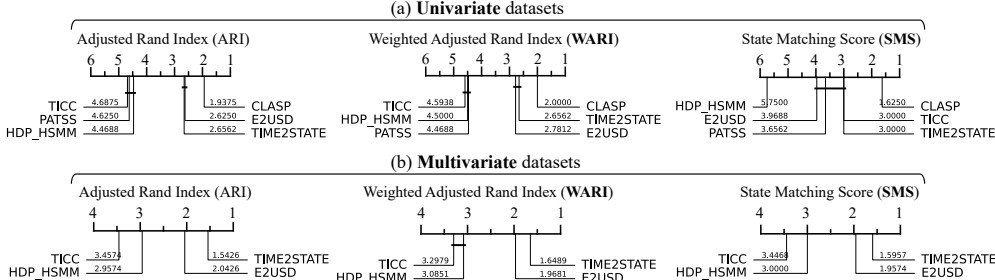

Figure 7: Critical diagrams of state detection algorithms on (a) multivariate and (b) univariate datasets.

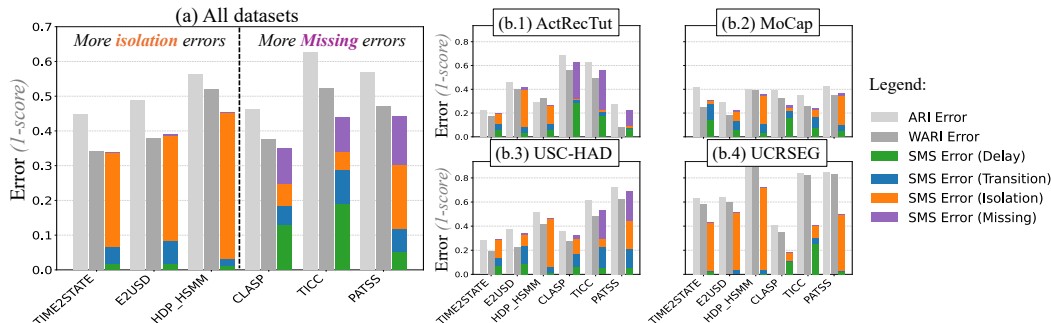

Figure 8: Error rate and error type contribution for (a) all datasets, and (b) per datasets.

outperforms other methods with a statistically significant difference in all measures. We also observe that TICC and HDP-HSMM are often ranked last in both univariate and multivariate settings.

**What is new:** While previous studies typically conclude at the level of the analysis described above, we employ SMS to further explore performance comparisons across different types of errors. Fig. 8 depicts the errors (i.e., $1-$Score) of each method, per dataset and measure, with the types of errors highlighted in the SMS bar. On average (Fig. 8(a)), we observe a significant distinction between (i) neural and probabilistic methods (such as Time2State, E2USD, and HDP-HSMM) which tend to produce more *isolated errors*, while (ii) other methods (such as ClaSP, TICC, and PaTSS) predominantly make *missing* and *delay* errors. However, the frequency and type of error vary significantly across datasets and methods, suggesting deep heterogeneity in segmentation behavior.

**What is ahead:** Beyond evaluation, the interpretable SMS framework suggests interesting research directions. The diversity in error types highlights opportunities for refining method selection, parameter tuning, and algorithm development. Specifically, error-type analysis guide the learning process and assess the parameter tuning step. For instance, the prevalence of isolated errors in Time2State, E2USD, and HDP-HSMM might be mitigated by adjusting parameters that control the number of clusters (e.g., the concentration parameter in the Dirichlet Process Gaussian Mixture Model used by Time2State). Conversely, for methods like ClaSP, TICC, and PATSS, which tend to produce missing errors, increasing the number of generated clusters could be beneficial (e.g., for ClaSP, by lowering the statistical test threshold for change point detection). Overall, error-type analysis can enhance both evaluation pipelines, as well as training, tuning and development processes.

## 5 Conclusion

We address a key gap in time series segmentation evaluation by formalizing a typology of four distinct errors types, and proposing a set of desirable properties for evaluation measures. We introduce two new evaluation measures, **WARI** and **SMS**, that overcome major limitations of existing approaches and provide important novel insights. Such insights, open promising directions for error-aware model selection, development, tuning, and ensembling. Overall, this work contributes to interpretable, robust, and customizable tools to advance the evaluation and design of segmentation algorithms.

## Acknowledgments and Disclosure of Funding

This work was funded in part by the French government under management of Agence Nationale de la Recherche (ANR) as part of the "France 2030" program, reference ANR-23-IACL-0008 (PR[AI]RIE-PSAI). The authors are grateful to the CLEPS infrastructure from the Inria of Paris for providing resources and support.

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

# A  Technical Appendices and Supplementary Material

**State Detection Related Work**

The Time2State method utilizes an encoder with a custom loss for a representation learning part, then clusters the time series using sliding windows. The E2USD algorithm, builds upon Time2State by targeting the high computational overhead hindering streaming applications and improves contrastive learning by better handling false negatives. HDP-HSMM adopts a non-parametric Bayesian approach to model temporal dependencies and duration distributions, while TICC utilizes temporal consistency and clustering to identify recurring patterns. The ClaSP + k-means approach relies on the state-of-the-art change point detection algorithm ClaSP, followed by kMeans clustering for grouping similar segments. Finally, PaTSS leverages pattern matching techniques to detect and align state transitions.

**Normalized Mutual Information (NMI)**

The NMI measures the mutual information $I(U; V)$ between two segmentations $U$ and $V$, normalized by their entropies $H(U)$ and $H(V)$:

$$\text{NMI}(U, V) = \frac{2I(U; V)}{H(U) + H(V)}$$

Mutual information $I(U; V)$ and entropies $H(U), H(V)$ are typically computed from the contingency matrix $n_{ij}$. The NMI ranges from 0 to 1, with 1 indicating perfect agreement between the two segmentations.

**Weighted Normalized Mutual Information (WNMI)**

Similar to the ARI, the NMI treats all points equally, regardless of their position within a segment. To incorporate temporal structure and boundary awareness, we propose a weighted version, the *Weighted Normalized Mutual Information* (WNMI).

Using the same weighting scheme as for WARI, where each observation $x_k$ has a weight $w_k$ based on its distance to the nearest boundary, we adapt the NMI calculation using the weighted contingency matrix $n_{ij}$.

**Datasets**

The datasets used in this study are publicly available and can be accessed through the following links:

- **PAMAP2** The Physical Activity Monitoring dataset is a comprehensive collection of data aimed at facilitating research in human activity recognition. It comprises recordings from nine subjects, each equipped with three Inertial Measurement Units (IMUs) placed on the wrist of the dominant arm, chest, and ankle, along with a heart rate monitor. The dataset includes 18 different physical activities, such as walking, running, and various household tasks. [15]

- **USC-HAD** The University of Southern California Human Activity Dataset is designed to support research in human activity recognition using wearable sensors. It contains data from 14 subjects, wearing a single MotionNode sensor on the front right hip. The sensor captures tri-axial accelerometer and gyroscope data at a sampling rate of 100 Hz. The dataset encompasses 12 activity classes, including walking, running, sitting, standing, and various transitional movements. [34]

- **UCR-SEG** The UCR Time Series Archive [35] is a repository of time-series datasets widely used for evaluating algorithms in various domains, including human activity recognition. It offers a diverse collection of datasets with varying lengths and dimensions, encompassing a range of activities and sensor modalities.

- **ActRecTut** This dataset is designed to support research in human activity recognition using body-worn inertial sensors [36]. It focuses on recognizing various hand gestures by analyzing data from inertial measurement units (IMUs) attached to the upper and lower arms. The dataset provides a comprehensive framework for designing and evaluating activity recognition systems, detailing each component and offering best practices developed by the research community.

- **MoCap** The CMU Graphics Lab Motion Capture Database [37] is a comprehensive collection of motion capture recordings performed by over 140 subjects. It includes a wide range of activities such as walking, dancing, and various sports, providing free motion data for research purposes. The database offers downloadable motion files in various formats, supporting research in fields like computer graphics, animation, and human motion analysis.

The properties of the datasets used are detailed in Table 1, as described in [20, 21].

Table 1: Properties of the datasets used in the experiments.

| Datasets | # States | # Channels | Length (k) | # Time series | # Segments | State duration (k) |
|----------|----------|------------|------------|---------------|------------|--------------------|
| MoCap    | 5~8      | 4          | 4.6~10.6   | 9             | 6~11       | 0.4~2.0            |
| ActRecTut| 6        | 10         | 31.4~32.6  | 2             | 42         | 0.02~5.1           |
| PAMAP2   | 11       | 9          | 253~408    | 10            | 18~25      | 2.0~40.3           |
| USC-HAD  | 12       | 6          | 25.4~56.3  | 70            | 12         | 0.6~13.5           |
| UCR-SEG  | 2~3      | 1          | 2~40       | 32            | 2~3        | 1~25               |

Table 2: Licenses of datasets used in our experiments.

| Dataset | License / Usage Terms |
|---------|-----------------------|
| PAMAP2  | CC BY 4.0 |
| USC-HAD | License not found; encouraged for use by ubiquitous computing researchers |
| UCR-SEG | License not found; widely used in research with at least 1000 papers citing the archive |
| ActRecTut | License not found |
| MoCap   | Free for research; commercial use allowed in products, but resale (even converted) is prohibited |

**Experimental Setup**

We evaluated each algorithm on each dataset, using the same hyperparameters as in [20]. We evaluated the runtime and performance of each algorithm, comparing their results on the different datasets. The experiments were conducted on a standard hardware setup including an Intel Core i7 processor and 32GB of RAM. We set a time limit of 24 hours for each dataset.

PaTSS and ClaSP did not run on PAMAP2, which contains sequences of 300,000 points long on average—about ten times longer than other datasets (see Table 1). They exceeded the runtime or memory limitations of our evaluation setup. While prior works reported results for these methods on subsampled versions of PAMAP2, we argue that evaluating on truncated sequences significantly distorts the segmentation task.

**Hyperparameters**

Table 3 provides the hyperparameters used in the experiments.

Table 3: Parameters for State Detection and Evaluation Measures

| Component | Parameters |
|---|---|
| State Detection Methods Parameters | Provided as `config.json` files in the `config` folder at the repository root. |
| Evaluation Measure Parameters | |

- **WARI Weight:**
  - $\alpha = 0.1$
- **SMS Weights:**
  - $w_{\text{delay}} = 0.1$
  - $w_{\text{transition}} = 0.3$
  - $w_{\text{isolation}} = 0.8$
  - $w_{\text{missing}} = 0.5$

Table 4: Algorithm performance across datasets and measures (mean $\pm$ std dev). 'x' indicates timeout or memory error.

| Dataset | Measure | HDP-HSMM | E2USD | PaTSS | Time2State | TICC | ClaSP |
|---|---|---|---|---|---|---|---|
| ActRecTut | F1 | $0.03 \pm 0.00$ | $0.03 \pm 0.01$ | $0.06 \pm 0.01$ | $0.05 \pm 0.01$ | $0.07 \pm 0.00$ | $0.08 \pm 0.00$ |
| | Covering | $0.27 \pm 0.00$ | $0.49 \pm 0.07$ | $0.66 \pm 0.25$ | $0.74 \pm 0.10$ | $0.33 \pm 0.01$ | $0.41 \pm 0.03$ |
| | ARI | $0.71 \pm 0.14$ | $0.54 \pm 0.09$ | $0.73 \pm 0.25$ | $0.78 \pm 0.09$ | $0.37 \pm 0.00$ | $0.31 \pm 0.05$ |
| | NMI | $0.70 \pm 0.08$ | $0.64 \pm 0.04$ | $0.74 \pm 0.16$ | $0.73 \pm 0.05$ | $0.53 \pm 0.00$ | $0.43 \pm 0.01$ |
| | WARI | $0.67 \pm 0.24$ | $0.60 \pm 0.17$ | $0.92 \pm 0.11$ | $0.83 \pm 0.18$ | $0.51 \pm 0.01$ | $0.44 \pm 0.04$ |
| | WNMI | $0.60 \pm 0.12$ | $0.56 \pm 0.04$ | $0.76 \pm 0.10$ | $0.68 \pm 0.05$ | $0.62 \pm 0.01$ | $0.47 \pm 0.01$ |
| | SMS | $0.74 \pm 0.11$ | $0.58 \pm 0.11$ | $0.77 \pm 0.22$ | $0.80 \pm 0.08$ | $0.42 \pm 0.01$ | $0.35 \pm 0.10$ |
| MoCap | F1 | $0.15 \pm 0.05$ | $0.18 \pm 0.06$ | $0.14 \pm 0.12$ | $0.19 \pm 0.04$ | $0.21 \pm 0.07$ | $0.25 \pm 0.05$ |
| | Covering | $0.53 \pm 0.11$ | $0.75 \pm 0.14$ | $0.50 \pm 0.25$ | $0.65 \pm 0.08$ | $0.74 \pm 0.16$ | $0.72 \pm 0.15$ |
| | ARI | $0.60 \pm 0.13$ | $0.71 \pm 0.18$ | $0.57 \pm 0.20$ | $0.58 \pm 0.15$ | $0.65 \pm 0.26$ | $0.61 \pm 0.16$ |
| | NMI | $0.70 \pm 0.09$ | $0.73 \pm 0.14$ | $0.68 \pm 0.14$ | $0.66 \pm 0.12$ | $0.68 \pm 0.26$ | $0.71 \pm 0.13$ |
| | WARI | $0.61 \pm 0.14$ | $0.82 \pm 0.20$ | $0.65 \pm 0.21$ | $0.75 \pm 0.14$ | $0.75 \pm 0.29$ | $0.68 \pm 0.19$ |
| | WNMI | $0.73 \pm 0.09$ | $0.77 \pm 0.12$ | $0.72 \pm 0.11$ | $0.70 \pm 0.10$ | $0.72 \pm 0.27$ | $0.74 \pm 0.12$ |
| | SMS | $0.64 \pm 0.12$ | $0.77 \pm 0.13$ | $0.63 \pm 0.17$ | $0.70 \pm 0.09$ | $0.76 \pm 0.15$ | $0.73 \pm 0.11$ |
| UCRSEG | F1 | $0.16 \pm 0.10$ | $0.18 \pm 0.10$ | $0.05 \pm 0.05$ | $0.22 \pm 0.19$ | $0.54 \pm 0.11$ | $0.59 \pm 0.10$ |
| | Covering | $0.14 \pm 0.12$ | $0.41 \pm 0.24$ | $0.20 \pm 0.21$ | $0.44 \pm 0.29$ | $0.67 \pm 0.20$ | $0.79 \pm 0.19$ |
| | ARI | $0.11 \pm 0.12$ | $0.36 \pm 0.23$ | $0.15 \pm 0.22$ | $0.37 \pm 0.30$ | $0.16 \pm 0.30$ | $0.59 \pm 0.33$ |
| | NMI | $0.20 \pm 0.18$ | $0.43 \pm 0.19$ | $0.17 \pm 0.21$ | $0.40 \pm 0.27$ | $0.17 \pm 0.30$ | $0.62 \pm 0.29$ |
| | WARI | $0.11 \pm 0.12$ | $0.40 \pm 0.28$ | $0.17 \pm 0.25$ | $0.42 \pm 0.34$ | $0.18 \pm 0.33$ | $0.65 \pm 0.36$ |
| | WNMI | $0.18 \pm 0.18$ | $0.41 \pm 0.21$ | $0.18 \pm 0.23$ | $0.40 \pm 0.28$ | $0.14 \pm 0.25$ | $0.59 \pm 0.33$ |
| | SMS | $0.28 \pm 0.11$ | $0.48 \pm 0.22$ | $0.50 \pm 0.16$ | $0.57 \pm 0.24$ | $0.60 \pm 0.18$ | $0.82 \pm 0.17$ |
| USC-HAD | F1 | $0.08 \pm 0.03$ | $0.14 \pm 0.04$ | $0.08 \pm 0.06$ | $0.09 \pm 0.04$ | $0.14 \pm 0.03$ | $0.15 \pm 0.03$ |
| | Covering | $0.18 \pm 0.04$ | $0.70 \pm 0.10$ | $0.49 \pm 0.17$ | $0.66 \pm 0.08$ | $0.60 \pm 0.10$ | $0.70 \pm 0.08$ |
| | ARI | $0.48 \pm 0.09$ | $0.63 \pm 0.11$ | $0.27 \pm 0.16$ | $0.72 \pm 0.10$ | $0.38 \pm 0.16$ | $0.64 \pm 0.12$ |
| | NMI | $0.71 \pm 0.06$ | $0.79 \pm 0.05$ | $0.50 \pm 0.16$ | $0.83 \pm 0.04$ | $0.69 \pm 0.10$ | $0.81 \pm 0.06$ |
| | WARI | $0.58 \pm 0.13$ | $0.77 \pm 0.13$ | $0.37 \pm 0.21$ | $0.81 \pm 0.12$ | $0.52 \pm 0.21$ | $0.73 \pm 0.15$ |
| | WNMI | $0.66 \pm 0.07$ | $0.75 \pm 0.06$ | $0.49 \pm 0.17$ | $0.77 \pm 0.05$ | $0.65 \pm 0.11$ | $0.75 \pm 0.06$ |
| | SMS | $0.54 \pm 0.08$ | $0.66 \pm 0.09$ | $0.28 \pm 0.19$ | $0.71 \pm 0.07$ | $0.44 \pm 0.14$ | $0.67 \pm 0.10$ |
| PAMAP2 | F1 | $0.02 \pm 0.04$ | $0.02 \pm 0.03$ | x | $0.03 \pm 0.04$ | $0.14 \pm 0.16$ | x |
| | Covering | $0.05 \pm 0.04$ | $0.43 \pm 0.08$ | x | $0.37 \pm 0.06$ | $0.51 \pm 0.16$ | x |
| | ARI | $0.28 \pm 0.07$ | $0.32 \pm 0.10$ | x | $0.31 \pm 0.10$ | $0.29 \pm 0.14$ | x |
| | NMI | $0.52 \pm 0.09$ | $0.60 \pm 0.10$ | x | $0.59 \pm 0.11$ | $0.55 \pm 0.07$ | x |
| | WARI | $0.42 \pm 0.16$ | $0.50 \pm 0.20$ | x | $0.49 \pm 0.20$ | $0.43 \pm 0.20$ | x |
| | WNMI | $0.60 \pm 0.18$ | $0.65 \pm 0.19$ | x | $0.65 \pm 0.19$ | $0.66 \pm 0.14$ | x |
| | SMS | $0.53 \pm 0.07$ | $0.54 \pm 0.16$ | x | $0.53 \pm 0.15$ | $0.52 \pm 0.16$ | x |

## Metrics Implementation

The ARI and NMI are calculated using the `sklearn` library in Python, which provides efficient implementations of these measures. The F1-score and covering scores are computed using a custom implementation, adapted from TSSB code.

## WARI vs. ARI

The link between WARI and ARI is displayed in Fig. 9.

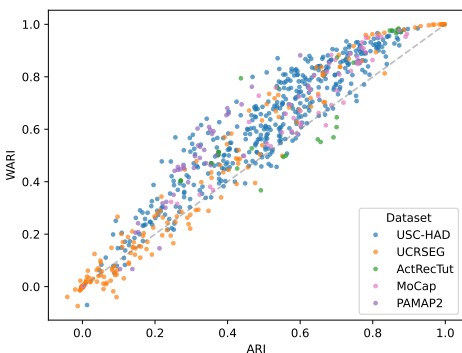

Figure 9: WARI against ARI score of each dataset.

## Critical Diagram for Change Point Detection

Figure 10 depicts the critical diagrams for F1 score and Covering score for both univariate and multivariate time series datasets.

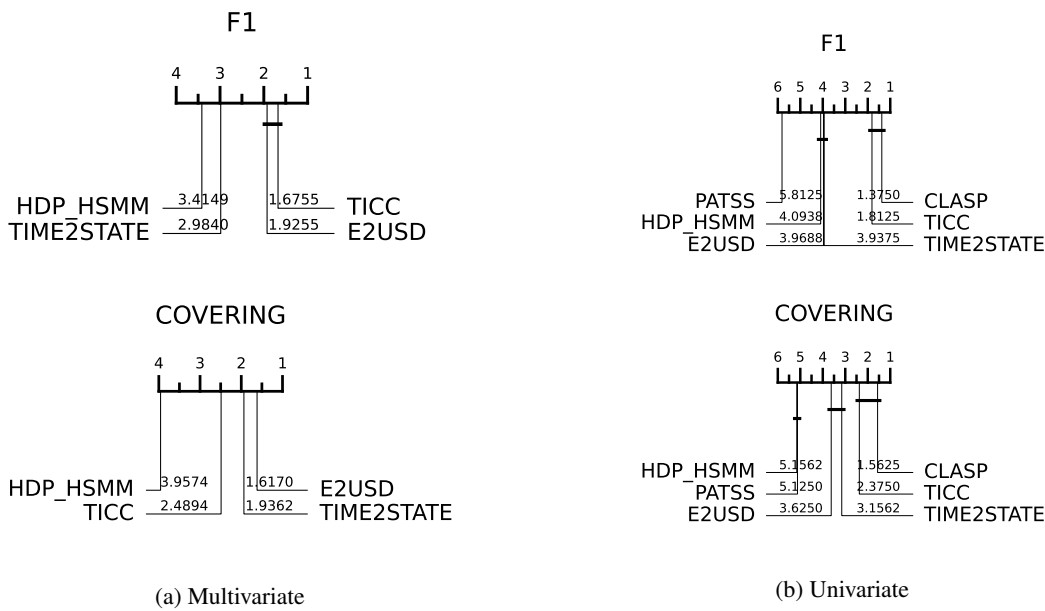

(a) Multivariate

(b) Univariate

Figure 10: Critical diagrams of state detection algorithms used as change point detectors. (a) Multivariate results. (b) Univariate results.

## SMS Robustness Evaluation

As shown in Algorithm 2, SMS incorporates penalty weights for different error types, allowing for customization to specific application requirements. For instance, in scenarios where reaction time is

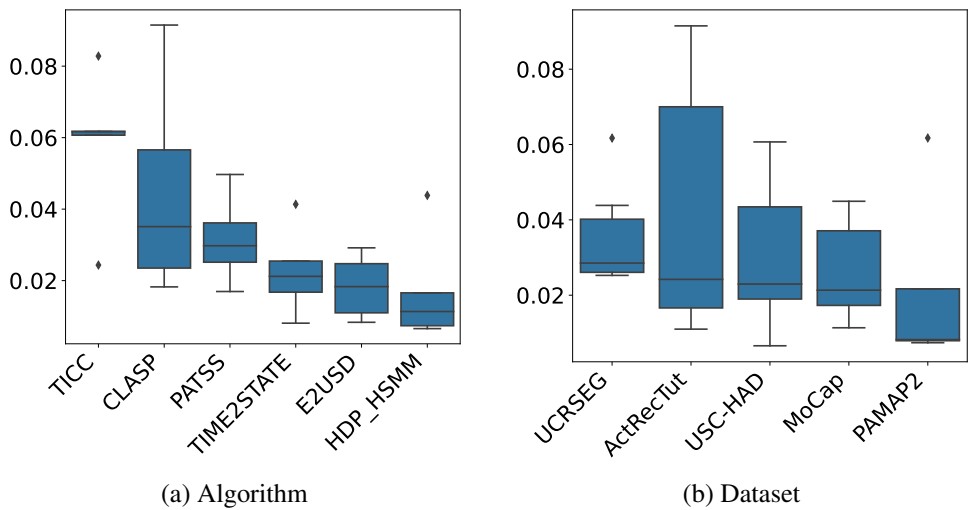

(a) Algorithm                                        (b) Dataset

Figure 11: SMS variability (std) across 100 runs with random uniform penalty weights in $[0, 1]$

critical and false positives are tolerable, delays might be penalized more heavily than missing states. Despite this flexibility, the SMS exhibits robustness to the specific choice of these penalty weights. The overall score is primarily influenced by the total number of errors rather than the precise weight distribution.

To demonstrate this, we evaluated all segmentations across the five datasets and six algorithms, using randomly assigned weights for each error type (drawn uniformly from $[0, 1]$ over 100 runs). As depicted in Fig. 11, the resulting score distributions showed limited variability, with an average standard deviation of $0.03169$. This indicates that while parameter tuning can refine the distinction between error types, the fundamental performance ranking remains largely consistent.

**Quantitative Evaluation of Error Types**

The count of each error type for each method across datasets using the SMS as evaluation measure is displayed in Fig. 12.

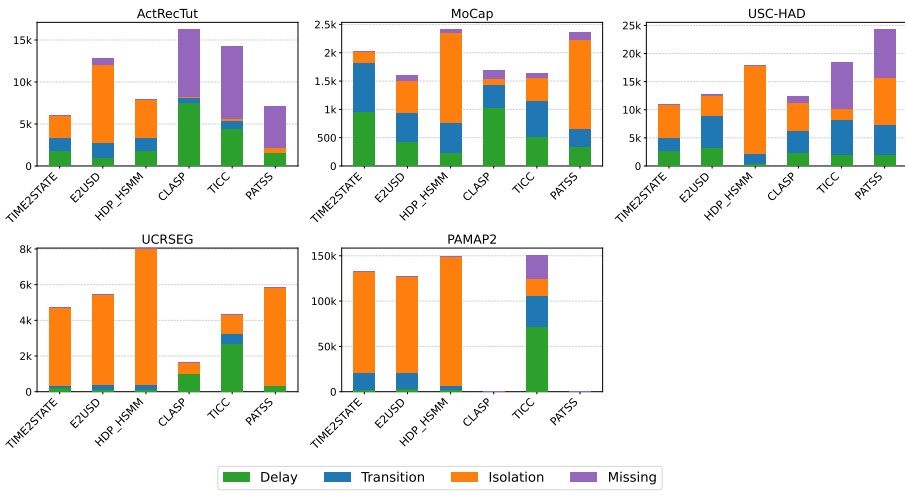

Figure 12: Count of each error type for each method across datasets using the SMS as evaluation measure

