# OpenReview forum: "Toward Interpretable Evaluation Measures for Time Series Segmentation"
_NeurIPS.cc/2025/Conference — NeurIPS 2025 poster_

### Official Review · Reviewer_rSwh · 2025-06-20

**Clarity:** 3
**Significance:** 3
**Originality:** 2
**Rating:** 5
**Confidence:** 5

**Summary:**

The paper introduces two novel measures the Weighted Adjusted Rand Index (WARI), enhances ARI by weighting segmentation errors based on their distance to true change points, improving sensitivity to temporal position; and the State Matching Score (SMS) a fine-grained measure that introduces a novel error typology (delay, isolation, transition, missing) and scores segmentation based on error type, position, and severity, allowing for customization and interpretability. These measures overcome the following limitations:
1. Change point-based measures do not adequately capture the overall quality of the segmentation itself.
2. Most widely used measures, such as Adjusted Rand Index (ARI), are point-based and thus treat all errors equally, failing to distinguish between different types of errors.
3. Current evaluation measures cannot track and categorize the nature of errors (e.g., delay vs. isolation), leading to limited interpretability.
The paper empirically demonstrates that WARI and SMS outperform traditional measures in offering nuanced and actionable insights across synthetic and real-world datasets.

**Questions:**

1.	The typology assumes clear distinctions between isolation, transition, and missing. But these boundaries can blur under label noise or gradual transitions. Is this typology robust to subjective labeling or soft state transitions?
2.	SMS depends on penalty weights and boundary distance parameters. How sensitive are results to these choices? Essentially how does hyperparameter tuning affects evaluation?
3.	If a segmentation algorithm is tuned with SMS in the loop, is it possible to "overfit" to SMS-friendly error distributions (e.g., producing short segments to minimize missing errors)?
4.	Can WARI be extended or adapted to domains outside time series (e.g., video segmentation)? i.e. is it generalizable?
5.	SMS involves Hungarian matching and error block scoring, potentially costly on long sequences or large batches. What are the real-world latency and memory footprints for high-frequency data streams?
6.	Real-world tasks often involve hierarchies or multi-scale segmentations (e.g., micro vs. macro activities). How do WARI and SMS handle hierarchical segmentations where ground truth may exist at different granularities?

**Ethical Concerns:**

["NO or VERY MINOR ethics concerns only"]

**Final Justification:**

I have read authors' response to my comments. I have also considered their response to concerns raised by fellow reviewers. While some concerns still remain, I think the contribution is important, post clarification.

**Limitations:**

Following are the limitations of the paper beyond author’s self-identified ones :-
1.	Computational Overhead and scalability concerns (In real time or high-resolution segmentation scenarios)
2.	Parameter Sensitivity and Subjectivity
3.	Lack of multi-scale support. The metrics assume a single-layer segmentation. They do not model segmentations where states can overlap or nest, which is common in real-world time series.
4.	Without normalization for random agreement (as ARI and WARI do), SMS may reward trivial segmentations. A model predicting only one state could get a deceptively high SMS under certain weight configurations.
5.	The paper evaluates segmentation in activity recognition and multivariate sensors. Generalizability to other domains remains untested.

**Quality:**

3

**Strengths And Weaknesses:**

Strengths:
1.	The paper offers a thorough critique of ARI, F1 and covering scores.
2.	The paper introduces a taxonomy of error types – delay, isolation, transition and missing – is a key conceptual contribution and add semantic depth, drawing a line between minor misalignment and severe structural failure.
3.	WARI is an elegant extension: it keeps the combinatorial structure of ARI, but adds a spatial awareness term, linking classification error with temporal topology. This weighting scheme can also be adapted to other point-based indices (as shown for NMI in appendix).
4.	SMS lets practitioners assign penalties to different error types. This makes the metric application-aware and useful in domain-specific scenarios.
5.	The evaluation covers a wide array of datasets and segmentation algorithms, showing consistent benefits of WARI and SMS across both synthetic and real-world data.
6.	Traditional metrics like F1 or ARI depend on binary pair agreement. SMS instead scores over structured intervals, enabling continuous error gradients.
Weakness:
1.	No theoretical guarantees. While empirical evidence is strong, no formal guarantees are given on the properties of WARI/SMS (e.g., bounds, consistency, sensitivity). The measures are heuristically motivated, not axiomatized.
2.	Subjective Configuration of SMS. It depends on user-defined penalty weights for error types. Without a data-driven framework to derive these weights, scores can be arbitrary across studies.
3.	SMS is sensitive to class imbalance and vulnerable to inflated scores when the number of predicted segments is low. This is not true for ARI and WARI as they are chance-corrected measures.
4.	Segmentation algorithms could be tuned specifically to perform well on SMS (e.g., avoiding missing errors), possibly gaming the metric rather than improving true generalization.

---

> ### Author Rebuttal · Authors · 2025-07-31
>
> > **The typology assumes clear distinctions between isolation, transition, and missing. But these boundaries can blur under label noise or gradual transitions. Is this typology robust to subjective labeling or soft state transitions?**
> >
> > We acknowledge that ground truth labeling, typically performed by domain experts, can be subjective, especially in cases of gradual transitions. Our error typology is designed to be robust to such scenarios. Subjective or noisy boundary placements are generally captured as delay errors. As argued in the paper (Section 2.2.), these can often considered as less severe than major structural errors (e.g., missing or transition errors). Therefore, in most applications, users can assign a lower penalty to these delay errors, making the method less sensitive against subjective/gradual transitions. Conversely, in domains where precise boundaries are critical, one would expect highly accurate labels, and SMS allows for a correspondingly higher penalty on delays.
>
> > **SMS depends on penalty weights and boundary distance parameters. How sensitive are results to these choices? Essentially how does hyperparameter tuning affects evaluation?**
> >
> > Thank you for raising this concern. SMS indeed depends on user-defined parameters, providing some flexibility for application-specific evaluation. However, the boundary-distance terms are normalized and do not involve any user-defined hyperparameter, avoiding complexity and keeping the weights independent, so they do not interfere with one another, making the tuning more intuitive.
> > Moreover, the influence of the penalty weights on the final score is bounded and predictable. Formally, the SMS score is bounded by $1 - \frac{(1+w_{\max})E}{N} \le \text{SMS} \le 1 - \frac{E}{N}$, where $N$ is the time series length, $E$ is the total length of all erroneous segments, and $w_{\max}$ is the maximum user-defined weight. This shows that the score is fundamentally anchored by the segmentation accuracy, with the weights providing a controlled adjustment.
> >Furthermore, we can demonstrate that the metric's sensitivity to changes in the weight configuration is directly proportional to the total error size $E$.
> > Indeed, the absolute change in the SMS score, for two sets of weights $w$ and $w'$, is bounded by $\lVert w - w' \rVert_\infty \cdot \tfrac{E}{N}$, where $\lVert w - w' \rVert_\infty$ is the maximum change across all weights. This property ensures that for a reasonably accurate segmentation (i.e., small $\tfrac{E}{N}$), the choice of weights has a limited and bounded impact on the score, preventing arbitrary fluctuations and ensuring that the evaluation remains robust. This aspect has been verified experimentaly and detailed in the robustness analysis in the appendix (p. 14), showing that the impact of these parameters on the evaluation remains limited.
>
> > **If a segmentation algorithm is tuned with SMS in the loop, is it possible to "overfit" to SMS-friendly error distributions (e.g., producing short segments to minimize missing errors)?**
> >
> > This is a valid concern. While it is theoretically possible for an algorithm to adapt its behavior based on SMS penalties, it is important to note that the dominant component of the penalty for any error type is its size, as detailed above. Therefore, an algorithm cannot significantly improve its SMS score without fundamentally reducing the overall size of its errors. Our robustness analysis (Appendix, p. 14) demonstrates that the score is primarily driven by this factor, ensuring a strong baseline of evaluation.
> >
> > The user-defined weights for specific error types act as a secondary, fine-tuning mechanism. We view this not as "overfitting" but as a form of guided optimization, where the metric's flexibility allows practitioners to steer models toward application-specific priorities. For example, if avoiding "missing" errors is critical for a task, SMS can be configured to penalize them heavily, encouraging the algorithm to prioritize their reduction. The practical integration of SMS into the training loop for such targeted optimization is discussed as a key area for future work in Section 4.2.
>
> > **Can WARI be extended or adapted to domains outside time series (e.g., video segmentation)? i.e. is it generalizable?**
> >
> >While our experiments focused on traditional time series data, the proposed metrics are designed to be generalizable. WARI and SMS operate on the temporal structure of segmentations—that is, a sequence of labels over time—and are agnostic to the underlying data modality.
> >
> >Consequently, for tasks involving temporal video segmentation, such as identifying scenes or actions, our metrics are directly applicable since a video can be treated as a sequence of frames (high dimensional multivariate time series). However, for spatial segmentation within individual video frames, which is not a one-dimensional temporal problem, these metrics would not apply. In essence, as long as the problem involves comparing a predicted temporal segmentation to a ground truth, our framework is suitable.
>
> > **SMS involves Hungarian matching and error block scoring, potentially costly on long sequences or large batches. What are the real-world latency and memory footprints for high-frequency data streams?**
> >
> > The computational complexity of SMS is determined by two main components: state alignment and error scoring. With `k_pred, k_true` the predicted and real number of clusters in the segmentation of a time series of length `T`:
> >
> > 1.  **State Alignment**: This involves building a `k_pred x k_true` cost matrix, which has a complexity of `O(k_pred * k_true * T)`, followed by running the Hungarian algorithm, which is `O(k³)` where `k = max(k_pred, k_true)`.
> > 2.  **Error Scoring**: This requires a single pass over the time series, resulting in a linear complexity of `O(T)`.
> >
> > The total complexity is `O(k_pred * k_true * T + k³)`. Since the number of states `k` is typically small and constant in most applications, the complexity is effectively linear with respect to the time series length `T`. The memory footprint is also linear, `O(T)`.
> >
> > In practice, the computational cost of SMS was negligible in our experiments compared to model training. Its linear scalability makes it efficient for offline analysis, even on long sequences or high-frequency data.
>
> > **Real-world tasks often involve hierarchies or multi-scale segmentations (e.g., micro vs. macro activities). How do WARI and SMS handle hierarchical segmentations where ground truth may exist at different granularities?**
> >
> > This is an interesting question that points to a more complex and distinct challenge in segmentation evaluation. Our current work focuses on "flat" or single-layer segmentation, where both the ground truth and the prediction exist at a single level of granularity.
> >
> > Consequently, WARI and SMS are not designed to handle hierarchical or multi-scale segmentations. Evaluating such structures would require a different conceptual framework capable of assessing agreement across multiple levels of abstraction simultaneously. This is a known limitation in the field, as highlighted in recent work such as the ClaSP paper (Ermshaus et al., 2023). To our knowledge, no established framework or real-world dataset currently exists for this specific task, with the exception of PaTSS (Carpentier et al., 2024), providing probabilitstic segmentation, but relying on synthetic dataset, and traditional measures for evaluation.
> >
> > We agree that developing metrics for hierarchical segmentation is an important direction for future research, but it is beyond the scope of the current paper.

---

> > ### Comment · Reviewer_rSwh · 2025-08-03
> > **Author response**
> >
> > I had missed Appendix A while reading the manuscript. The response addresses most of my concerns while the concerns about generalizability and sensitivity could be addressed in future work. I believe the paper is an important contribution in TS segmentation. I'm therefore happy to raise my score.

---

### Official Review · Reviewer_kyud · 2025-06-24

**Clarity:** 4
**Significance:** 3
**Originality:** 3
**Rating:** 4
**Confidence:** 5

**Summary:**

This paper introduces two metrics for evaluating the quality of time series segmentations, addressing key limitations of existing evaluation methods, namely their inability to capture segmentation quality, differentiate between error types, or provide meaningful interpretability.
The first metric, WARI (Weighted Adjusted Rand Index), extends the traditional Adjusted Rand Index by incorporating the temporal positions of segmentation errors, enabling it to distinguish between early, late, or localized mismatches. The second metric, SMS (State Matching Score), identifies and scores four types of segmentation errors (delay, isolation, transition, and missing segments) and allows users to assign application-specific weights to each error type. The proposed metrics are empirically validated across multiple datasets and segmentation methods, comparing their diagnostic power to existing evaluation metrics.

**Questions:**

1. Hungarian matching applicability: The use of Hungarian matching in the evaluation framework appears well-suited for self-supervised or unsupervised segmentation settings, where the predicted states may not align with ground-truth labels. However, for supervised segmentation methods that learn a direct mapping to specific state labels, is this matching still necessary? Could it distort the evaluation if exact label alignment is already learned?

2. Comparison of SMS and ARI scores: Figure 6 presents a clear and helpful visualization of how SMS compares to ARI. However, it consistently shows SMS scores as higher than ARI, which raises questions about potential score calibration or scaling differences. Is this overestimation expected? How should it be interpreted, especially since in other figures (e.g., Figure 8), SMS appears lower than ARI in many cases? Clarifying this would help with understanding how to read and compare these scores across experiments.

**Ethical Concerns:**

["NO or VERY MINOR ethics concerns only"]

**Final Justification:**

I believe this paper is tackling and interesting and important problem. Overall, the paper was well-written and easy to follow. I originally had some concerns, but the rebuttal was very helpful in resolving those.

**Limitations:**

While the paper does not explicitly discuss limitations, I believe it's important to address a main concern, which is the sensitivity of the proposed metrics to user-defined weighting variables and weights. One of the strengths of this work is its potential to become a standard evaluation metric that others can use to benchmark and compare segmentation methods. However, the flexibility to customize weights for different error types ($\alpha$ in WARI and error type weights in SMS), while valuable for application-specific analysis, also opens the door to overfitting the metric to favor particular methods.

This raises a concern: if weighting is tuned to benefit a specific model or use case, comparisons across methods may become less meaningful. In such cases, the metric's value may shift more toward being a diagnostic tool rather than a consistent benchmarking standard. A deeper discussion of this trade-off, and perhaps guidance on default weight settings or best practices for fair comparison, would be a valuable addition to the paper.

**Paper Formatting Concerns:**

No formatting concerns

**Quality:**

3

**Strengths And Weaknesses:**

Strengths:

1. The paper thoroughly analyzes the problem of evaluating segmentation in time series and clearly articulates the properties that distinguish good from bad segmentations. This results in a detailed and structured framework for analyzing segmentation quality.
2. I believe this work has strong potential, especially given the lack of robust existing solutions, and it could ultimately become a standard benchmark metric for time series segmentation.
3. The visuals, result descriptions, and analysis are excellent. The use of diverse examples is particularly effective in motivating the problem and demonstrating how different metrics respond to segmentation errors.


Weaknesses:

1. My major concern is that the proposed scores—especially SMS—are highly sensitive to the weighting scheme. While this flexibility is useful for application-specific evaluation, it becomes a downside when the metric is used for benchmarking. There should be at least some guidelines or constraints on how weights are selected. For example, do weights exhibit linear behavior—i.e., if "delay" is twice as important as "missing segment," should its weight simply be doubled? Even with that assumption, the choice of weights still feels somewhat arbitrary. The paper would benefit from a sensitivity analysis or guidance on recommended weighting strategies for common use cases.
2. To establish these metrics as state-of-the-art tools in the field, the paper would benefit from a broader set of experiments. While the selected datasets and models are solid, I recommend comparisons against basic metrics such as error rate or MSE, as well as sensitivity analyses for the proposed metrics. One particularly compelling experiment would be to show that using the proposed metric for model selection leads to better downstream performance. This would demonstrate that the metric not only captures meaningful differences but is also useful in practice—especially since much of the current analysis remains qualitative or relative to metrics with known limitations.
3. Typo in algorithm: Line 8 in Algorithm 2 assigns the same weights to all components. I assume these are intended to be different weights, so this should be corrected.

---

> ### Author Rebuttal · Authors · 2025-07-31
>
> > **Hungarian matching applicability: The use of Hungarian matching in the evaluation framework appears well-suited for self-supervised or unsupervised segmentation settings, where the predicted states may not align with ground-truth labels. However, for supervised segmentation methods that learn a direct mapping to specific state labels, is this matching still necessary? Could it distort the evaluation if exact label alignment is already learned?**
> >
> > Thank you for the insightful question. Indeed, in a fully supervised setting, where a model is trained to predict specific, pre-defined state labels, the Hungarian matching step is not necessary. Our framework can be easily adapted for this scenario by bypassing the state alignment step. This is achieved by modifying a single line in Algorithm 2 (Line 1 is replaced by $\tilde{P} \gets P$), which ensures that the predicted labels are directly compared against the ground-truth labels. The rest of our formalism, including the error typology and weighting scheme, remains fully applicable and provides the same diagnostic power in both supervised and unsupervised settings.
>
> > **Comparison of SMS and ARI scores: Figure 6 presents a clear and helpful visualization of how SMS compares to ARI. However, it consistently shows SMS scores as higher than ARI, which raises questions about potential score calibration or scaling differences. Is this overestimation expected? How should it be interpreted, especially since in other figures (e.g., Figure 8), SMS appears lower than ARI in many cases? Clarifying this would help with understanding how to read and compare these scores across experiments.**
> >
> > Thank you for this keen observation. The two metrics operate on fundamentally different principles. ARI is a combinatorial, pair-counting metric where a single misclassified point can disproportionately impact the score by affecting its relationship with all other points ($N-1$ pairs). In contrast, SMS is a segment-based metric calculated as $1 - (E + P) / N$, where $N$ is the time series length, $E$ is the total length of all erroneous segments, and $P$ is the sum of penalties applied to these segments. Its score is anchored by the proportion of correctly classified points, with nuanced adjustments based on the type, size, and context of error segments. This design means that the impact of an error on the SMS score is more nuanced as localized and semantically weighted. Consequently, SMS often assigns a less severe penalty for minor temporal misalignments compared to ARI's stricter, binary approach, leading to the expected higher scores observed in Figure 6.
> >
> > Regarding Figure 8, please note that the y-axis represents the error rate (calculated as 1 - score), where a lower value indicates better performance. Therefore, the observation that SMS is often lower than ARI in Figure 8 actually signifies that SMS assigns a *better* score, which is consistent with the trend shown in Figure 6. We will clarify the y-axis label in the final version to prevent any confusion.
>
> > **My major concern is that the proposed scores—especially SMS—are highly sensitive to the weighting scheme. While this flexibility is useful for application-specific evaluation, it becomes a downside when the metric is used for benchmarking. There should be at least some guidelines or constraints on how weights are selected. For example, do weights exhibit linear behavior—i.e., if "delay" is twice as important as "missing segment," should its weight simply be doubled? Even with that assumption, the choice of weights still feels somewhat arbitrary. The paper would benefit from a sensitivity analysis or guidance on recommended weighting strategies for common use cases.**
> >
> > Thank you for raising this important concern. We acknowledge the trade-off between the flexibility of user-defined weights for application-specific diagnostics and the need for standardization in benchmarking.
> >
> > To answer your specific question: yes, the weights are designed to have a direct, linear impact on the penalty. The total penalty `P` is a weighted sum of individual error penalties. Therefore, if a "delay" error is considered twice as important as a "missing segment" error, its weight should indeed be doubled. This linear relationship is an intended feature, designed to make the tuning process intuitive.
> >
> > While the flexibility might seem arbitrary, the influence of these weights is formally bounded and predictable. The SMS score is fundamentally anchored by segmentation accuracy (`1 - E/N`, where `E` is the total error size and `N` is the series length), with the weights providing a controlled adjustment. Given SMS expression, the score is bounded by $1 - \frac{(1+w_{\max})E}{N} \le \text{SMS} \le 1 - \frac{E}{N}$, where $w_{\max}$ is the maximum weight. Furthermore, the metric's sensitivity to changes in the weight configuration is directly proportional to the total error size `E`. Specifically, the absolute change in the SMS score for two weight sets, `w` and `w'`, is bounded by $\lVert w - w' \rVert_\infty \cdot (E/N)$. This property ensures that for reasonably accurate segmentations, the choice of weights has a limited and bounded impact, preventing arbitrary fluctuations. A detailed robustness analysis is provided in the appendix (p. 14).
> >
> > Regarding guidelines, we believe the best approach is to empower practitioners to make informed choices. SMS provides a detailed diagnostic breakdown, showing the size, type, and position of each error along with its associated penalty. This transparency allows users to observe the direct impact of their weighting scheme and tune it to reflect the priorities of their specific use case, moving the process from arbitrary to application-driven.
>
> > **To establish these metrics as state-of-the-art tools in the field, the paper would benefit from a broader set of experiments. While the selected datasets and models are solid, I recommend comparisons against basic metrics such as error rate or MSE, as well as sensitivity analyses for the proposed metrics. One particularly compelling experiment would be to show that using the proposed metric for model selection leads to better downstream performance. This would demonstrate that the metric not only captures meaningful differences but is also useful in practice—especially since much of the current analysis remains qualitative or relative to metrics with known limitations.**
> >
> > Thank you for these constructive suggestions, which help clarify the scope and utility of our work. We agree that comparing against basic metrics is important for contextualizing our contributions.
> >
> > Regarding error rate (or its complement, accuracy), this is implicitly included in our framework. As detailed in our response to your other concern about weight sensitivity, SMS simplifies to `1 - error_rate` when all weights are set to zero. This provides a clear and interpretable baseline for our metric.
> Mean Squared Error (MSE) is not directly applicable to this task. The integer values of state labels are categorical and arbitrary; they do not carry numerical meaning that would make an MSE calculation valid. Even after the Hungarian alignment, the matched labels (e.g., `true_label=2`, `pred_label=3`) have no inherent numerical relationship. Converting the labels to a binary one-hot representation and computing MSE would be equivalent to calculating the point-wise error rate.
> Regarding the sensitivity analysis, we have addressed this in detail in our response to your other question about the weighting scheme and in the appendix (p. 14). We provide formal bounds on the metric's sensitivity to weight changes, ensuring that for reasonably accurate models, the score remains stable and predictable.
> > Regarding the suggestion to demonstrate improved downstream performance through model selection, our paper already provides the foundation for this. As shown in Figures 7 and 8, SMS offers a detailed error diagnostic that allows practitioners to identify the best-performing algorithm for a given dataset based on application-specific error priorities. This diagnostic can be used as-is for model selection. While a full experimental validation of the downstream impact is a key focus of our ongoing work, we suggest that current results already establish the practical utility of SMS for this purpose.
>
> > **Typo in algorithm: Line 8 in Algorithm 2 assigns the same weights to all components. I assume these are intended to be different weights, so this should be corrected.**
> >
> > Thank you for pointing out this ambiguity. Line 8 of Algorithm 2 uses the weight $w_e$, where the subscript $e$ denotes the error type identified in Line 7 (i.e., delay, isolation, transition, or missing). The notation is intended to select the specific weight corresponding to the given error type. We will revise the notations to make this more explicit and prevent any misinterpretation.

---

> > ### Comment · Reviewer_kyud · 2025-08-03
> > **Rebuttal response**
> >
> > Thank you for addressing my concerns. The rebuttal has been very informative. Please make sure to include the discussion on weights interpretation and selection in the updated manuscript.

---

### Official Review · Reviewer_fvnY · 2025-06-29

**Clarity:** 3
**Significance:** 3
**Originality:** 2
**Rating:** 4
**Confidence:** 4

**Summary:**

This paper proposed two evaluation measures, WARI and SMS for time series segmentation.
Compared with existing approaches, the proposed method can model the length, positions, and types of error.

**Questions:**

1. Have you considered formalizing theoretical properties of SMS and WARI, such as consistency or convergence guarantees?

2. How does WARI fundamentally differ from other distance-sensitive clustering metrics? Are there scenarios where WARI behaves non-trivially compared to ARI with pre-alignment?

3. Can you provide examples or experiments where using SMS as a selection criterion leads to improved model performance or better real-world segmentation?

4. Could SMS or WARI be integrated into model training objectives or hyperparameter tuning workflows?

**Ethical Concerns:**

["NO or VERY MINOR ethics concerns only"]

**Final Justification:**

I agree with the response of authors on my concerns for improving this work.

**Limitations:**

1. SMS is not chance-adjusted, which may lead to misleadingly high scores in trivial segmentations.

2. The choice of penalty weights in SMS can influence results, although some robustness is demonstrated.

3. Lack of model-guided improvement loop: it remains unclear whether these metrics can be used to improve segmentation algorithms in practice.

**Quality:**

2

**Strengths And Weaknesses:**

### [Strengths]
1. The motivation of this work is clear. It sounds reasonable that four desired properties can help more meaningful evaluation of time series segmentation.
2. The organization of paper is clear and the writing is easy to understand.

### [Weaknesses]
1. The paper lacks deep theoretical analysis or formal guarantees for WARI and SMS (e.g., monotonicity, consistency, convergence).
2. WARI is a relatively straightforward modification of ARI with time-based weights, which may be viewed as incremental rather than fundamentally novel.
3. While the proposed metrics diagnose segmentation quality, it is unclear how they lead to improved model performance or better real-world decision-making.

---

> ### Author Rebuttal · Authors · 2025-07-31
>
> > **Have you considered formalizing theoretical properties of SMS and WARI, such as consistency or convergence guarantees?**
> >
> > Thank you for raising this important point. We can indeed establish several formal properties for both WARI and SMS that guarantee their consistency and bounded behavior.
> >
> > **WARI Properties:**
> > WARI is a principled extension of the Adjusted Rand Index (ARI). It inherits all of ARI's fundamental properties, including adjustment for chance and a fixed range (typically close to 0 for random agreement and 1 for perfect agreement). WARI's behavior is controlled by a single parameter, $\alpha$, which modulates the weight assigned to each data point based on its distance to the nearest true segment boundary.
> >
> > -   **Consistency with ARI:** When $\alpha=0$, all points are weighted equally, and WARI behaves as the standard ARI.
> > -   **Asymptotic Convergence:** As $\alpha \to \infty$, the weight of each point becomes dominated by its distance from the boundary. We can show that the $\alpha$ terms cancel out in the WARI formula, causing the score to converge to a stable value. This limit corresponds to an ARI computed using a contingency matrix where each point's contribution is weighted purely by its distance to the nearest boundary. This ensures that the metric's behavior is stable and predictable even for large values of $\alpha$. We have observed this convergence in practice, with the WARI score stabilizing for large $\alpha$.
> >
> > **SMS Properties:**
> > We have established several formal properties for SMS that guarantee its robustness and bounded behavior.
> >
> > According to its definition in Algorithm 2, SMS is calculated as $\text{SMS} = 1 - (E+P)/N$, where $N$ is the time series length, $E$ is the total length of all erroneous segments, and $P$ is the sum of penalties applied to these segments. The penalty term $P$ is a function of user-defined weights, allowing for application-specific adjustments.
> >
> > -   **Interpretability and Baseline:** When all weights are set to zero, the penalty term $P$ vanishes, and SMS simplifies to $1 - E/N$, which is equivalent to segmentation accuracy (the fraction of correctly labeled points). This provides a clear and interpretable baseline for the score.
> > -   **Boundedness and Stability:** The influence of the penalty weights on the final score is bounded and predictable. Formally, the SMS score is bounded by $1 - \frac{(1+w_{\max})E}{N} \le \text{SMS} \le 1 - \frac{E}{N}$, where $w_{\max}$ is the maximum user-defined weight. This shows that the score is fundamentally anchored by the segmentation accuracy, with the weights providing a controlled adjustment.
> > -   **Robustness to Weight Selection:** We can demonstrate that the metric's sensitivity to changes in the weight configuration is directly proportional to the total error size $E$. Specifically, the absolute change in the SMS score for two weight vectors, $w$ and $w'$, is bounded by $\lVert w - w' \rVert_\infty \cdot (E/N)$. This property ensures that for a reasonably accurate segmentation (i.e., small $E/N$), the choice of weights has a limited and bounded impact on the score, preventing arbitrary fluctuations and ensuring that the evaluation remains robust. An experimental evaluation for these properties is provided in the appendix (p.14)
> > We will add a detailed discussion of these theoretical properties for both WARI and SMS, along with their formal proofs, to the appendix of the revised manuscript.
>
> > **How does WARI fundamentally differ from other distance-sensitive clustering metrics? Are there scenarios where WARI behaves non-trivially compared to ARI with pre-alignment?**
> >
> > Thank you for this insightful question. To the best of our knowledge, no distance-sensitive metrics are available for evaluating time series segmentation. Our literature review shows that even recent state-of-the-art methods (e.g., Time2State (Wang et al., 2023), E2USD (Lai et al., 2024), and the recent CLaP preprint (Ermshaus et al., 2025)) rely on standard, non-distance-sensitive metrics like ARI, AMI, and NMI. A key challenge in this context is that predicted state labels are arbitrary and do not align with ground-truth labels, making direct, point-wise distance calculations meaningless without a prior matching step. While metrics for time series clustering exist, they are not directly applicable to segmentation evaluation due to this label permutation issue. It is to fill this gap that we proposed WARI, as an adaptation of a non-distance-sensitive measure, and SMS, a native distance-based measure that relies on label pre-alignment.
> >
> > ARI is permutation-invariant, meaning its final score is independent of the specific numerical labels assigned to states. Therefore ARI provides the same score with or without pre-alignment, and we can refer to the examples provided in the paper to answer your specific question. We provide an example of such a scenario, where WARI behaves non-trivially compared to ARI, in Figure 5, which is discussed in detail in Section 4.1. In this example, we present two non-trivial segmentations, S1 and S2. While S2 is qualitatively superior to S1 (based on the desirable properties P1-P4 outlined in Section 2.2), the standard ARI metric incorrectly assigns a higher score to S1. In contrast, our proposed WARI metric correctly favors S2, demonstrating its ability to capture nuanced temporal errors that ARI overlooks.
>
> > **Can you provide examples or experiments where using SMS as a selection criterion leads to improved model performance or better real-world segmentation?**
> >
> > Thank you for this question, pointing towards metric's practical utility. While we do not present a dedicated experiment on downstream task performance, our paper provides the foundation and examples for how SMS can be used as a selection criterion to achieve better real-world segmentation outcomes.
> >
> > The primary utility of SMS in model selection comes from its detailed error diagnostic. Unlike monolithic scores, SMS provides a transparent breakdown of every error, detailing its type, size, position, and associated penalty. This diagnostic power allows a practitioner to move beyond a simple "which model is better?" to "how is this model better, and do its strengths align with my application's needs?"
> >
> > For instance, Figures 7 and 8 in our paper serve as a clear example of this. They provide a detailed breakdown of algorithm performance, highlighting the different error profiles each method exhibits. A practitioner can use this diagnostic to select the algorithm that best aligns with their specific requirements. For example, in a medical application where failing to detect a critical state (a "missing" error) is far more dangerous than a slight timing misalignment (a "delay" error), a user could select the model that minimizes "missing" errors, even if its overall score is slightly lower than a competitor's. This choice, guided by the SMS diagnostic, directly leads to a better and safer real-world segmentation outcome.
> >
> > Furthermore, under an i.i.d. assumption, a practitioner could evaluate several algorithms on a small, labeled subset of their data. By analyzing the error types with SMS, they could then select the most suitable algorithm for their specific use case and apply it to the larger, unlabeled dataset, ensuring the chosen model's behavior is optimized for what matters most in their domain.
>
> > **Could SMS or WARI be integrated into model training objectives or hyperparameter tuning workflows?**
> >
> >This question highlights a key direction for future work, which we discuss in Section 4.2. The integration of SMS and WARI into model development workflows can be viewed in two distinct contexts:
> >
> >1.  **Supervised and Semi-Supervised Settings:** For tasks like model selection or hyperparameter tuning where a labeled validation set is available, both SMS and WARI can be directly used as evaluation criteria. As discussed in our response to the previous question, a practitioner can select the model or hyperparameters that yield the best SMS/WARI score on the labeled data, tailored to the specific error types relevant to their application. This is a straightforward and highly practical application of our metrics.
> >
> >2.  **Unsupervised Settings:** Integrating these metrics directly into the training objective of an unsupervised model (i.e., without a ground truth) is more challenging. Since SMS and WARI are extrinsic measures that require a reference segmentation, they cannot be used as a loss function in a purely unsupervised context. Adapting our framework for such scenarios is currently our focus. An option we are considering is to use surrogate internal metrics (derived from WARI and SMS), leveraging the model centrality hypothesis, as has been proposed for related time series tasks such as outlier detection (Ma et al., 2023) and anomaly detection (Li et al., 2022). Investigating the applicability of such unsupervised model selection frameworks to time series segmentation, within the WARI/SMS scope is a key direction of our future work.

---

> > ### Comment · Reviewer_fvnY · 2025-08-06
> >
> > I would like to thank the authors for your timely and clear responses to my comments.
> >
> > My rating is upgraded from 3 as 4.

---

### Official Review · Reviewer_nH5j · 2025-07-02

**Clarity:** 3
**Significance:** 2
**Originality:** 3
**Rating:** 4
**Confidence:** 4

**Summary:**

This paper introduces two evaluation measures, WARI (Weighted Adjusted Rand Index) and SMS (State Matching Score), that "provide a more accurate assessment of segmentation quality and uncover insights, such as error provenance and type, that are inaccessible with traditional measures."

**Questions:**

Please see weaknesses.

**Ethical Concerns:**

["NO or VERY MINOR ethics concerns only"]

**Limitations:**

Yes. The paper has provided a thorough discussion about its the limitations of its proposed methods.

**Quality:**

3

**Strengths And Weaknesses:**

Strengths:

The paper is well-written and logically structured, and the proposed metrics offer interpretable diagnostics by identifying and categorizing fundamental segmentation error types, which enhances comparative analysis across methods. The experimental results on both synthetic and real-world datasets further support the practical usefulness of the proposed metrics, particularly SMS, which allows error-type-aware evaluation and facilitates downstream model analysis and tuning.

 Weaknesses:

There are still some limitations to be addressed. First, the authors compare their proposed metrics only with traditional evaluation measures (e.g., ARI, F1, and Covering) but omit comparisons with recent advancements in segmentation evaluation. This omission weakens the empirical rigor of the study. Including such baselines would better contextualize the work within the current landscape of metric design. If directly comparable methods are unavailable, discussing related approaches and explicitly differentiating their scope and focus in the related work section would still be valuable.
Additionally, while the approach is practical and engineering-driven, it would benefit from deeper theoretical analysis or formal guarantees to strengthen the methodological foundation and generalizability of the proposed metrics.

---

> ### Author Rebuttal · Authors · 2025-07-30
>
> > **The authors compare their proposed metrics only with traditional evaluation measures (e.g., ARI, F1, and Covering) but omit comparisons with recent advancements in segmentation evaluation.**
> >
> >
> > Thank you for this insightful comment. A key motivation for our work is the observation that, despite significant advancements in segmentation algorithms, the methods for their evaluation have not evolved accordingly. Our literature review confirms that even the most recent state-of-the-art papers on time series segmentation rely exclusively on traditional metrics. For state detection, methods such as Time2State (Wang et al., 2023), E2USD (Lai et al., 2024), and the recent CLaP preprint (Ermshaus et al., 2025) use ARI, AMI, or NMI. Similarly, for change point detection, the most recent algorithms like ClaSP (Ermshaus et al., 2023) are evaluated using F1 and Covering. This highlights a critical and persistent gap in the field, which our work directly aims to address. We will clarify this point in the related work section to better contextualize our contribution. However, if we have overlooked recent advancements in evaluation metrics, we would be grateful if the reviewer could share the relevant references.
>
> > **If directly comparable methods are unavailable, discussing related approaches and explicitly differentiating their scope and focus in the related work section would still be valuable.**
> >
> > This is a valuable point. While we did not find direct counterparts to WARI and SMS for time series segmentation, we agree that discussing metrics from related domains could strengthen the paper. We will expand our related work section to include a discussion of evaluation metrics from other sequential data domains (e.g., anomaly detection) and differentiate their objectives and methodologies from our approach, thereby better situating our work within the broader landscape of evaluation metric design.
>
> > **While the approach is practical and engineering-driven, it would benefit from deeper theoretical analysis or formal guarantees.**
> >
> > Thank you for raising this important point. We can indeed establish several formal properties for SMS that guarantee its robustness and bounded behavior.
> >
> > According to its definition in Algorithm 2, SMS is calculated as $\text{SMS} = 1 - (E+P)/N$, where $N$ is the time series length, $E$ is the total length of all erroneous segments, and $P$ is the sum of penalties applied to these segments. The penalty term $P$ is a function of user-defined weights, allowing for application-specific adjustments. When all weights are set to zero, the penalty term vanishes, and SMS simplifies to $1 - E/N$, which is equivalent to segmentation accuracy (the fraction of correctly labeled points). This provides a clear and interpretable baseline for the score.
> >
> > The influence of the penalty weights on the final score is bounded and predictable. Formally, the SMS score is bounded by $1 - \frac{(1+w_{\max})E}{N} \le \text{SMS} \le 1 - \frac{E}{N}$, where $w_{\max}$ is the maximum user-defined weight. This shows that the score is fundamentally anchored by the segmentation accuracy, with the weights providing a controlled adjustment. Furthermore, we can demonstrate that the metric's sensitivity to changes in the weight configuration is directly proportional to the total error size $E$.
> >
> > Specifically, the absolute change in the SMS score is bounded by $\lVert w - w' \rVert_\infty \cdot \tfrac{E}{N}$, where $\lVert w - w' \rVert_\infty$ is the maximum change across all weights. This property ensures that for a reasonably accurate segmentation (i.e., small $\tfrac{E}{N}$), the choice of weights has a limited and bounded impact on the score, preventing arbitrary fluctuations and ensuring that the evaluation remains robust.
> >
> > A brief justification for these properties is as follows. Given $Err$ as the set of erroneous segments, we have $E=\sum_{e \in Err} |e|$ and $P= \sum_{e \in Err} |e|\, w_e \, \operatorname{context}(e)$, where both the weight $w_e$ and the context term $\operatorname{context}(e)$ are in $[0, 1]$. This leads to the bound $0 \le P \le w_{\max} E$. Substituting this into the SMS definition gives the above presented bounds.
> >
> > Regarding the sensitivity analysis, let $w$ and $w'$ be two weight vectors. The difference in scores is $|\text{SMS}(w) - \text{SMS}(w')| = |P(w') - P(w)|/N$. The change in total penalty is bounded by $|P(w) - P(w')| \le \sum_{e \in Err} |e| \cdot |w_e - w'_e|$
> >
> > $\le \lVert w - w' \rVert_\infty\sum_{e \in Err} |e| = \lVert w - w' \rVert_\infty E$. Combining these gives the sensitivity bound $|\text{SMS}(w) - \text{SMS}(w')| \le \lVert w - w' \rVert_\infty (E/N)$.

---

> > ### Comment · Reviewer_nH5j · 2025-08-06
> >
> > Thank the authors for the rebuttal. As my original scores were positive, I would like to maintain my scores.

---

### Decision · Program_Chairs · 2025-09-17

**Decision:**

Accept (poster)

**Comment:**

This paper introduces two novel evaluation measures, WARI (Weighted Adjusted Rand Index) and SMS (State Matching Score), to more accurately assess time series segmentation quality. WARI extends the traditional Adjusted Rand Index by weighting segmentation errors based on their temporal position, making it more sensitive to the timing of mismatches. The SMS metric provides a fine-grained analysis of errors, identifying and scoring four distinct error types: delay, isolation, transition, and missing segments. This approach offers detailed insights into error provenance and type, which is not possible with existing methods. The metrics allow for custom weighting of error types and have been empirically validated to demonstrate their diagnostic superiority.

The ARI is a widely used metric for clustering, so it is surprising that this paper successfully extends it to time series segmentation with WARI and SMS. Since robust evaluation metrics are crucial for developing better machine learning methods, this is likely a significant, long-term contribution to time series segmentation research. All the reviewers are happy about the paper, and I am also voting for acceptance.